# Kind of X and X of Interest: An Ontology Design Pattern to Reconcile Web of Thing Ontologies

Maxime Lefrançois[1,*,†], Catherine Roussey[2,‡], Fatma-Zohra Hannou[3,†], Victor Charpenay[4,†] and Antoine Zimmermann[5,†]

[1]Mines Saint-Etienne, Univ Clermont Auvergne, INP Clermont Auvergne, CNRS, UMR 6158 LIMOS, Saint-Étienne, France

[2]MISTEA, University of Montpellier, INRAE, Institut Agro, Montpellier, France

[3]EDF R&D, Palaiseau, France

## Abstract

Several Web of Thing (WoT) ontologies defined by different standard organizations share common objectives: describing measurements and possible actions. Unfortunately, their modelling seems quite similar but not easy to align. This article proposes the pattern "Kind of X and X of Interest". It conciliates two ways of modelling proposed in those ontologies to clarify their understanding and possibly integrate them. The paper illustrates the reuse of the pattern in the ontology of the CoSWoT project, which aims to extend the Semantic Web of Things (SWoT) to constrained devices. We also showcase possible generalization tracks to reuse the pattern in other projects addressing domains other than WoT, such as the European project Omega-X.

## Keywords

Semantic Web of Things, ontology description, Ontology design pattern

## 1. Introduction

The Web of Things (WoT) is expanding the Web to physical reality. That is, it allows us to access and manipulate physical entities via Web-based protocols. With WoT abstraction, one can directly GET the temperature of a place or the speed of a vehicle, rather than fetch a document that contains data to be interpreted as a measurement of the temperature; with WoT abstraction, one can move a robotic arm, rather than update a digital record that triggers a command to a robot's interface. In WoT, one important principle is that things must be semantically described so as to know what they are able to do, what they are doing, and the results of their actions (be it measurement, movement, turning on, etc.). Several ontologies exist for this purpose, including

*15th Workshop on Ontology Design and Patterns (WOP 2024), Colocated with the 22nd International Semantic Web Conference (ISWC 2024) November 12, 2024. Baltimore, USA*

[*]Corresponding author.

[†]These authors contributed equally.

✉ maxime.lefrancois@emse.fr (M. Lefrançois); catherine.roussey@inrae.fr (C. Roussey); fatma-zohra.hannou@edf.fr (F. Hannou); victor.charpenay@emse.fr (V. Charpenay); antoine.zimmermann@emse.fr (A. Zimmermann)

🌐 https://www.maxime-lefrancois.info/ (M. Lefrançois); https://sites.google.com/view/rousseycatherinephd/ (C. Roussey); https://www.vcharpenay.link/ (V. Charpenay); https://w3id.org/people/az/ (A. Zimmermann)

🆔 0000-0001-9814-8991 (M. Lefrançois); 0000-0002-3076-5499 (C. Roussey); 0000-0003-4747-1232 (F. Hannou); 0000-0002-9210-1583 (V. Charpenay); 0000-0003-1502-6986 (A. Zimmermann)

standard ones. Yet, even them have ambiguities in how they should be used. For instance, the class ssn:Property of the SSN ontology may have instances that are generic (e.g., *ambient temperature*), with which a specific observation can be specified, e.g. that a measurement is made for the ambient temperature of a specific place. Alternatively, one can assume that, e.g., *the ambient temperature of room 429* is itself an instance of a property, distinct from the ambient temperature of a different place. While SSN does not explicitly distinguish between the generic and specific properties, the latest version of SAREF has this distinction. The class saref:Property exists for the generic type, while saref:PropertyOfInterest serves for the specific one. Likewise, SAREF distinguish saref:FeatureKind as the class of generics associated with the specific saref:FeatureOfInterest. In order to unify these design approaches while keeping maximum flexibility of usage, we introduce here the "Kind of X and X of Interest" design pattern, where 'X' can be any aspect relevant to the WoT and even beyond. The paper is organized as follows: Section 2 presents the main ontologies that cover what our pattern is mostly concerned with. Section 3 presents the pattern itself. Section 4 demonstrates how the pattern was used in ontologies of a research project. Section 5 provides a use case illustrating practical use of the pattern to build knowledge graphs. An overview of existing uptake is provided in Section 6. Finally, Section 7 concludes the paper.

## 2. State Of The Art

Our design pattern was introduced to give a common ground to three partially complementary, partially competing standards on the Web of Things domain: SOSA/SSN, WoT Thing Descriptions, and SAREF. We present each of them in what follows.

### 2.1. SOSA/SSN

The ontologies for *Sensors, Observations, Samples, and Actuators* (SOSA) and *Semantic Sensor Networks* (SSN) are W3C standards for the descriptions of sensor devices and related systems and processes. They started their existence through the SSN Incubator Group that published its final report [1] and the first SSN ontology in 2011. At the same time, the Open Geospatial Consortium (OGC) published the UML model Observations and Measurements (O&M) V2.0 [2]. By 2014, SSN had been massively reused in datasets and by other ontologies, so the OGC and W3C Spatial Data on the Web joint Working Group (SDW-WG) worked on its standardization until 2017 when the recommendation was published with the dual ontology SOSA/SSN [3, 4, 5]. This version consists of a lightweight core module called SOSA, a more expressive SSN module, a separate SSN-Systems module describing system capabilities, and a set of alignment modules with other ontologies. The description of sensor contains the entity of interest and its characteristic it observes, the measurements it makes. A measurement description is called an observation. Analogously, the description of actuator contains the entity of interest and its property on which it can act, the actuations it makes and the result of these actuations.

As of 2024, the SDW-WG is working on a new version of SOSA/SSN[1] that should contain an extension for observation collections, and that will align with the V3.0 of the OGC standard

---

[1]https://github.com/w3c/sdw-sosa-ssn/

O&M now named *Observations, measurements and samples*).

## 2.2. Thing Description

*Web of Things (WoT) Thing Description* [6] (or TD for short) is a W3C standard for describing devices that expose Web-based interfaces to access their data and activate them. A *Thing* is defined in the WoT architecture [7] as the abstraction of an entity (virtual or physical) that can be manipulated by an IoT application, *e.g.* an object, a service, or a logical entity such as a room or a building. The **Thing Description** [8] ontology describes the *interaction affordances* of objects, i.e. the possibilities objects offer in terms of network communication to manipulate their state, or trigger a process execution. The td:Affordance class, as defined in TD, is in line with the preexisting ontology design pattern for affordances[2], in which an affordance is the reification of some statement 'X affords Y', where Y is a template for action (a Task, in the design pattern specification, or a request/response interaction, in TD). TD can be used in conjunction with the Hypermedia controls [9] ontology and the RDF vocabulary for JSON Schema [10] to precisely describe how to formulate a valid request (HTTP, CoAP, MQTT, etc.) to the object [11].

## 2.3. SAREF

In 2014, the European Telecommunications Standards Institute (ETSI) studied knowledge models related to IoT, and developed the Smart Applications Reference ontology (**SAREF**)[3] [12, 13]. SAREF today consists of a core module, and twelve extensions for application areas such as energy, buildings, or agriculture. Initially centered on the concept of saref:Device as a physical object fulfilling one or more functions, the SAREF ontology has been progressively improved over the versions. Version V3.2.1 of SAREF Core [14] was released in January 2024, and demonstrates a strong desire for convergence towards SOSA/SSN. For example, the saref:Measurement class is deprecated in favour of the saref:Observation class, taken from SOSA/SSN. SAREF modelling choices are suggested to the SDW-WG, and some have already been integrated such as the saref:ProcedureExecution class which generalizes the observations and actuations, and the saref:PropertyOfInterest class which describes the observable qualities intrinsically linked to an entity of interest. In addition, SAREF defines a set of reference ontology patterns [15] which constrain its use. These patterns indicate which classes and properties can or cannot be specialized, and in what context. A new major version (V4.1.1) of SAREF Core should be published in 2024 along with a new major version (V2.1.1) of each of the SAREF extensions.[4]

## 2.4. Different Modelling Choices

SOSA/SSN, TD, and SAREF ontologies have distinct governance and evolved independently. We note a desire of convergence in modelling practices and sharing some similar modelling questions (see [3]). For example, the question of whether an instance of the class ssn:Property should be generic (e.g. room temperature) or specific to an entity of interest (e.g. office

---

[2]https://odpa.github.io/patterns-repository/Affordance/Affordance
[3]https://saref.etsi.org/
[4]https://portal.etsi.org/tb.aspx?tbid=726#/50611-work-programme

temperature 429) has long been debated. There are uses for both ways of modelling.[5] The decision was made in SAREF to restrict the use of saref:Property only for generic properties, and to introduce saref:PropertyOfInterest to describe properties specific to an entity of interest. SOSA/SSN adopted this choice as well. Likewise, some users describe types of sensors with sosa:Sensor (e.g. DHT22), while others describe instances (e.g. the DHT22 sensor in the room 429). SAREF introduces saref:FeatureKind to describe archetypes of entities of interest, but not the saref:DeviceKind or saref:SensorKind subclass. The new modelling choice has been discussed but rejected in SOSA/SSN.[6]

Thus, sometimes the specific is defined by the suffix "*OfInterest*", sometimes the generic is defined by the suffix "*Kind*", making modelling decision difficult to understand.

Moreover, using SAREF and SOSA/SSN jointly with TD requires a clear distinction between the generic and the specific. According to the definition of interaction affordances, only specific instances of FeatureOfInterest, Device or Sensor may afford request/response interactions. Generic concepts or archetypes do indeed have no agency. Even though it may be possible to generically state that all temperature sensors afford temperature reading, such a statement is not consistent with TD, according to which what can be afforded is necessarily a specific interaction targeting a single Web host. If a class of objects encompasses both specific and generic entities, one looses the ability to detect possible inconsistent TD statements.

## 3. Pattern: Kind of X and X of Interest

To ease the understanding of the SAREF and TD modelling, and to integrate the SSN/SOSA one, we propose a design pattern that we call "*Kinds of X and X of Interest*", shown in Figure 1.

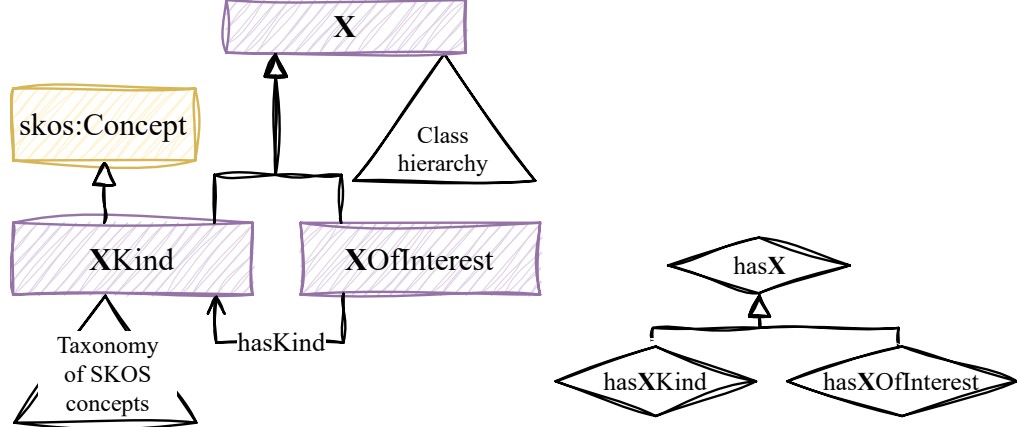

**Figure 1:** Architectural Design Pattern "*Kinds of X and X of Interest*", and its extension "*has X*"

The class **X** (e.g. Feature, Property, Device) represents the class of entities whose type is indeterminate (neither generic nor specific). It can be specialized in a class hierarchy. The class

---

[5]See https://github.com/w3c/sdw-sosa-ssn/issues/106
[6]See https://github.com/w3c/sdw-sosa-ssn/issues/107, https://github.com/w3c/sdw-sosa-ssn/issues/209

**X** is equivalent to the disjoint union of **X**Kind and **X**OfInterest, which represent respectively the class of archetypes of **X** and specific **X**. **X**Kind is a subclass of skos:Concept, and these instances are organized into a SKOS model using the skos:broader and skos:narrower properties. Local restrictions on **X**Kind force more specific and more generic concepts to also be of type **X**Kind.

The hasKind property binds a specific entity to its archetypes. A local restriction on **X**OfInterest forces any object of the hasKind property to be of archetype **X**Kind. Additionally, hasKind is a superproperty of the property chain hasKind ∘ skos:broader, so that an instance of **X**OfInterest "inherits" the more generic archetype **X**Kind. The hasKind property being "non-simple"[7] by this axiom, it should not be subject to cardinality restrictions. We can nevertheless add a local existential restriction **X**OfInterest⊑ ∃hasKind.**X**Kind, and a universal restriction **X**Kind⊑ ∀hasKind.⊥. This last restriction prohibits the use of hasKind on instances of type **X**Kind.

When a subclass of **X** is defined, it is possible, but not mandatory, to use the same architectural design pattern. For example, the classes Sensor, SensorKind, and SensorOfInterest, will have respectively subclasses Device, DeviceKind, and DeviceOfInterest.

Finally, it is common to model that an instance of the pattern for **Y** refers to an instance of the pattern for **X**. A **X**OfInterest will be specific to a unique **Y**OfInterest. For example, a property of interest is specific to an entity of interest. The pattern therefore proposes an extension "*has X*" which defines six properties: has**X**, is**X**Of, has**X**Kind, is**X**KindOf, has**X**OfInterest, and is**X**OfInterestOf, the latter being functional. Three axioms of type subPropertyChainOf allow us to infer the types:

$$\text{skos:broader} \circ \text{has}\mathbf{X}\text{Kind} \sqsubseteq \text{has}\mathbf{X}\text{Kind}$$
$$\text{hasKind} \circ \text{has}\mathbf{X}\text{Kind} \sqsubseteq \text{has}\mathbf{X}\text{Kind}$$
$$\text{has}\mathbf{X}\text{OfInterest} \circ \text{hasKind} \sqsubseteq \text{has}\mathbf{X}\text{Kind}$$

Even if has**X**Kind is non-simple, the class **X**OfInterest can receive a local restriction of cardinality = 1 on is**X**OfInterestOf without violating the global restrictions of OWL 2 DL.

This pattern facilitates the separation of concerns: ontology developers will favour the definition of **X** subclasses; thesauri developers or online catalogs managers will favour the instantiation of **X**Kind classes; application developers will favour the instantiation of **X**OfInterest classes.

## 4. CosWoT Ontology

The Constrained Semantic Web of Things (CoSWoT) project[8] addresses the bottlenecks linked to the use of the Semantic Web in Internet of Things Objects, especially those related to semantic interoperability. Thus, it proposes a dedicated ontology based on standard ones.

---

[7]Non-simple properties are subject to global restrictions to satisfy the OWL 2 DL profile.
[8]https://coswot.gitlab.io/

## 4.1. Feature

The class coswot:Feature describes entities representing any real-world entity having a property or state that will be observed or controlled. An instance of coswot:Feature is either an instance of coswot:FeatureOfInterest, i.e. a specific entity of the real world, or an instance of coswot:FeatureKind, i.e. an archetype entity. Figure 2 illustrates the pattern applied to coswot:Feature.

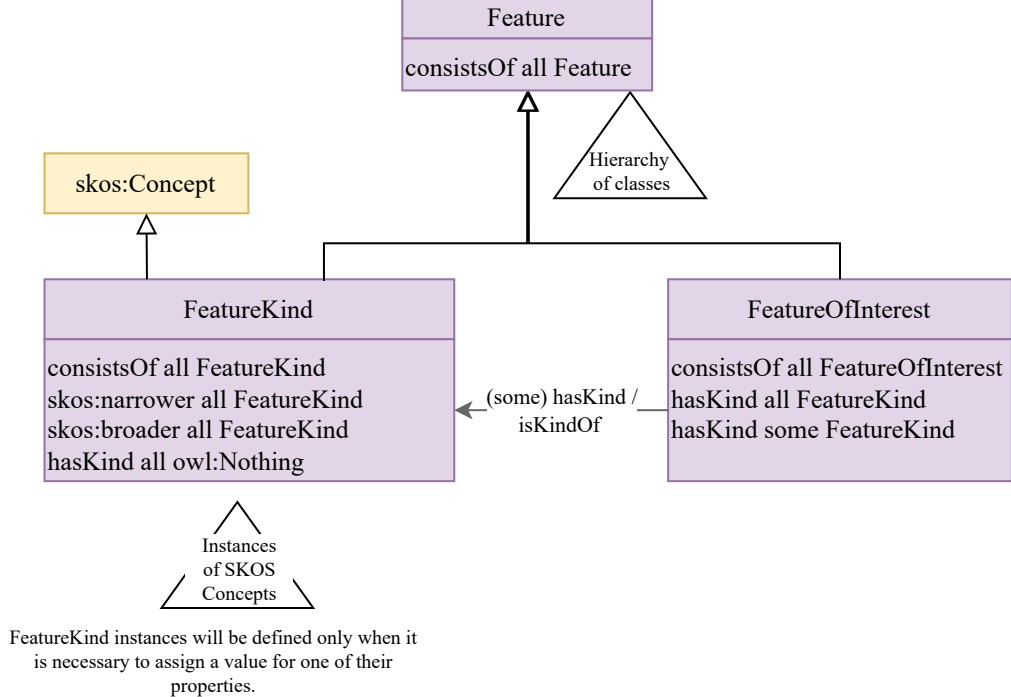

**Figure 2:** Chowlk Diagram of coswot:Feature class.

## 4.2. Property

The description of coswot:Property applies the pattern with the "*has X*" extension. A coswot:Property describes the properties of entities. Properties refer to the identifiable qualities of entities that devices can observe. An instance of coswot:Property is either an instance of coswot:PropertyOfInterest, i.e. a characteristic specific to an identified entity in the real world, or an instance of coswot:PropertyKind, i.e. a property archetype that can apply to different entities. Figure 3 illustrates the pattern applied to coswot:Property.

## 4.3. Device

The class coswot:Device describes devices, including sensors and actuators. Since devices are real-world entities, it specializes the coswot:Feature class. A representation of a device

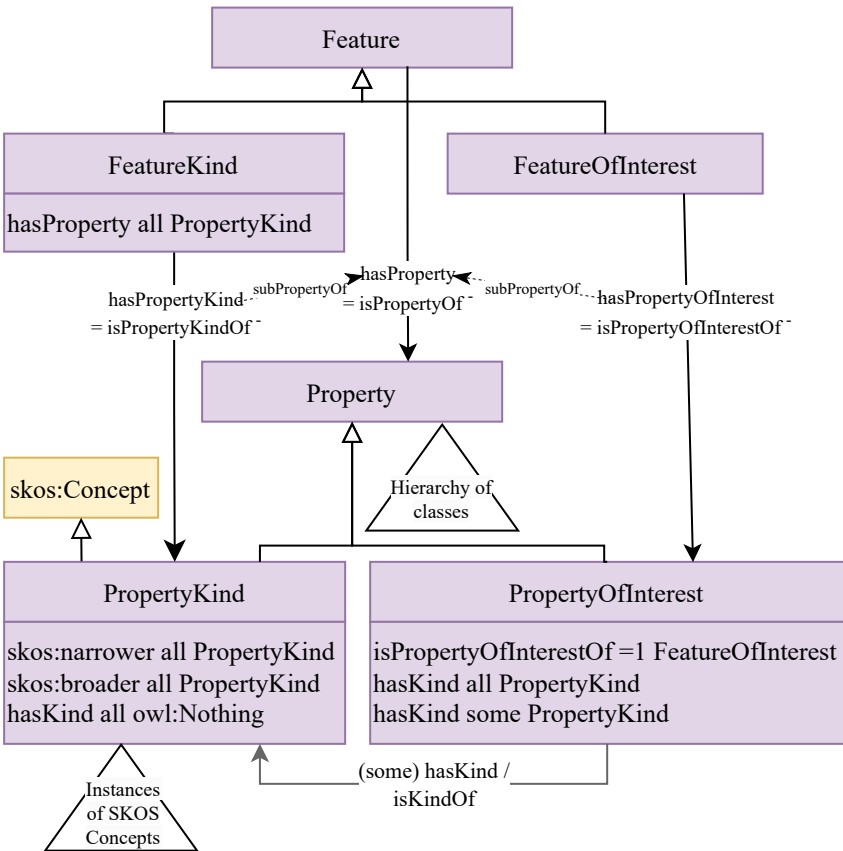

**Figure 3:** Chowlk Diagram of coswot:Property class

is an instance of coswot:Device, subclass of coswot:Feature. This representation is either an instance of coswot:DeviceOfInterest, i.e. an identified device from the real world, or an instance of coswot:DeviceKind, i.e. an archetype of devices. The class coswot:Sensor (or respectively coswot:Actuator) specializes the class coswot:Device.

An instance of coswot:SensorOfInterest identifies a sensor from the real world, e.g. the thermometer of an identified room. An instance of coswot:SensorKind identifies an archetype of sensors, e.g. 30 cm depth soil thermometer.

Parallel to sensors, an instance of coswot:ActuatorOfInterest identifies an actuator from the real world, e.g. the heater of an identified room. An instance of coswot:ActuatorKind identifies an archetype of actuators, e.g. 500 W auxiliary heater from Atlantic brand.

## 5. Use Case

The scope of the CoSWoT project extends to two application domains: digital agriculture, and smart buildings. We illustrate the separation of concerns that the "Kinds of X and X of Interest" architectural design pattern offers on an application in the smart building domain, specifically

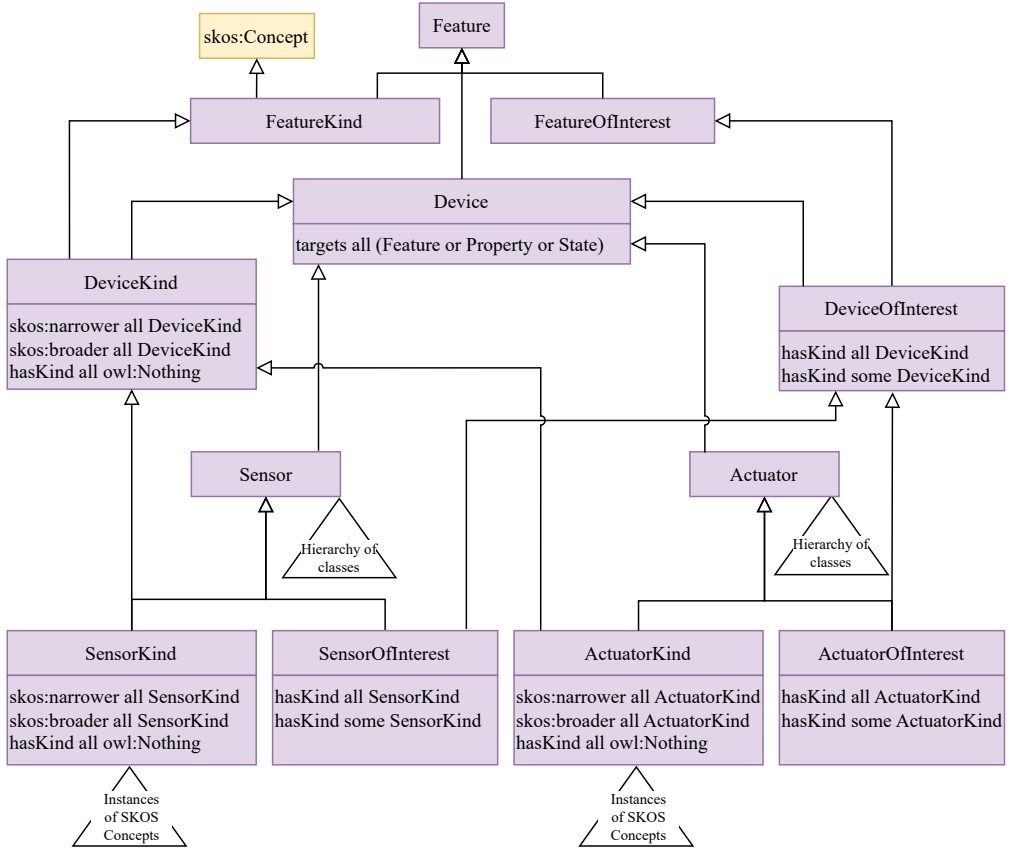

**Figure 4:** Chowlk Diagram of coswot:Device class

on the Espace Fauriel (EF) building of Mines Saint-Étienne [16].

Our example involves several devices deployed in office 429:

- Auxiliary heater: equipped with a thermometer, and remotely controllable;
- Window platform: located near the window, hosting a thermometer, a sensor that evaluates the air's carbon dioxide level ($CO_2$ sensor), an actuator arm that can open the window and provide its status. This platform holds hold communication, computation, and reasoning capabilities.
- Board platform: near the white board hosting a thermometer, a $CO_2$ sensor, and holding communication capabilities;
- Door platform: near the door with a thermometer, a $CO_2$ sensor, and communication capabilities.

In our example, the $CO_2$ sensors measure every 5 seconds, and the platforms send their measurements to the window platform to calculate the maximum $CO_2$ concentration. Based on this measurement, this platform will deduce the air quality of the office. The goal is to automate the window opening based on the air quality. A knowledge graph is created to support this application (See Figure 5).

In the context of the smart building domain, the first step toward using the coswot ontology is its specialization for smart building use cases. This involves the creation of a domain ontology, which lists the relevant subclasses for each module (class **X** in the pattern). Thus, the class coswot:Feature class is specialized into coswot:Room, coswot:OfficeRoom, coswot:Hall, etc. The class coswot:Property: is specialized into coswot:Temperature, coswot:CarbonDioxydeConcentrationInAir, coswot:AirQuality, etc. The class coswot:Sensor is specialized into coswot:TemperatureSensor, coswot:CarbonDioxideSensor, etc.

Next, online taxonomies and theasurus (e.g. SKOS models) provide hierarchies of feature kinds, property kinds, etc. For example the buildingSMART Data Dictionary (bSDD) defines building elements and properties as instances of bsdd:Class and bsdd:Property.[9] Using existing controlled vocabularies and terminologies enables interoperability with other resources, by aligning with what can be used in other similar applications. Instances are consequently reused as instances of **X**Kind, or domain-specific taxonomy is created with for example coswot:AirTemperature, coswot:AccurateSensor, etc.

Finally, instantiating the classes **X**ofInterest to enumerate real-world individuals relevant to application's scope is a task that is entrusted to the capable hands of application developers. Their role in this process is pivotal and their expertise is invaluable. In our example, the feature of interest studied is office 429 in the Espace Fauriel building. This office is unique; its characteristics are also, hence all devices hosted in the office are listed as individuals of class coswot:DeviceOfInterest. Similarly, the room's air temperature and CO2 concentration are instances of coswot:PropertyOfInterest.

Figure 5 displays an excerpt of the knowledge graph for the office 429 window's opening scenario.

The core of the CoSWoT ontology contains 8 instances of the pattern "*Kinds of X and X of Interest*", 4 having extension "*has X*". It is available at URL https://w3id.org/coswot/core/

## 6. Uptake

Apart from the CoSWoT project where it has been defined and applied, the pattern is already partly used in at least one other project, where the need to consideration separation has been identified in early ontology design phases.

The Omega-X project [10] is a European project that aims to create an energy data space to foster innovation and economic growth within the energy sector by facilitating secure and interoperable data sharing. It emphasizes adherence to FAIR data principles, ensuring that data and services are findable, accessible, interoperable, and reusable. The ontology developed for the Omega-X project [11] is modular, enabling to capture the complex relationships and interactions within the energy domain. It integrates existing domain standards and reference ontologies to ensure semantic interoperability and support developing and reusing datasets and services across the energy data space. The project considers four energy domain sub-domains, called

---

[9]See RDF representations of entities <class/IfcDoor> and <prop/Width> obtained through content negotiation with base https://identifier.buildingsmart.org/uri/buildingsmart/ifc/4.3/.

[10]https://omega-x.eu/

[11]https://w3id.org/omega-x/

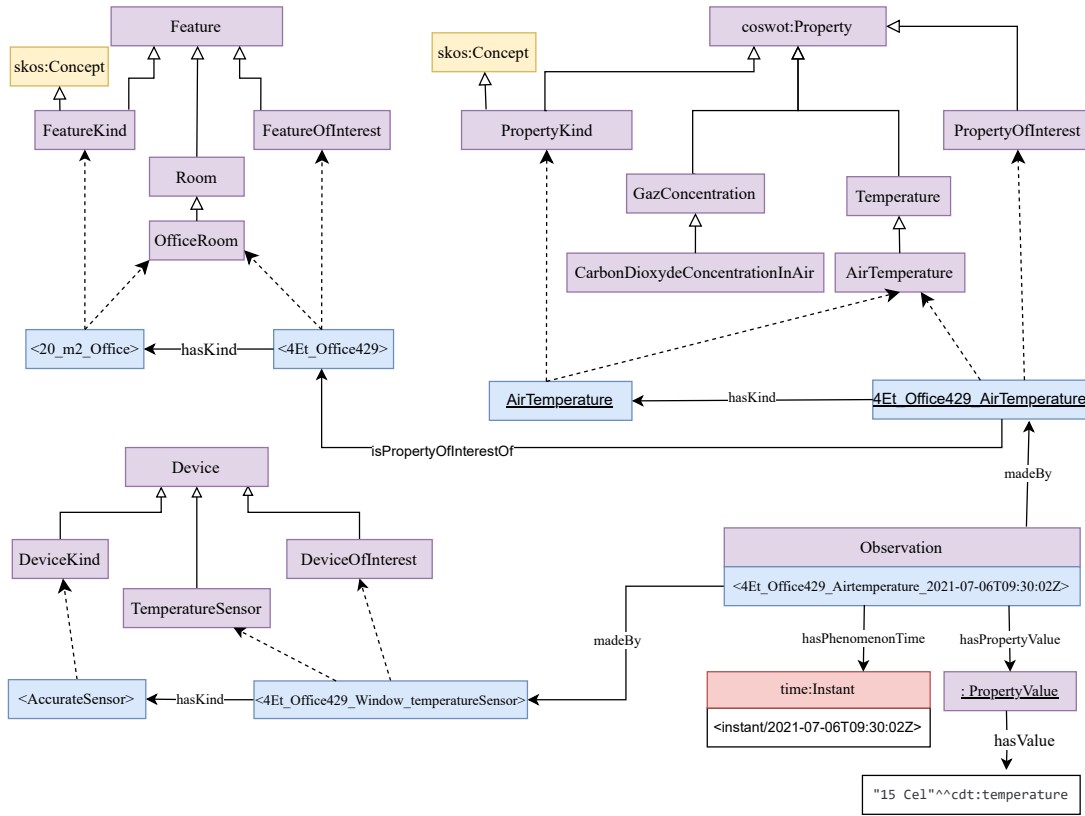

**Figure 5:** Excerpt of the knowledge graph for the office 429 window's opening scenario

use case families (Flexibility, Electromobility, Local Energy Communities, Renewables) Modules are organized into three hierarchical levels: Top-level, domain and use case families ontology modules.

In the frame of the Omega-X project, two modules were designed in the spirit of the "Kinds of X and X of Interest" pattern:

- **Property**: In the domain level, a property taxonomy (PropertyKind) is specialized into energy domain properties to cover ElectricityProduction, Voltage,... On the other hand, application knowledge graphs, produced by data providers indicate real-world individuals for properties VelverdeMeteoStationTemperature, MaiaSubstationVoltageLevel,...
- **Infrastructure**: In the domain level, a taxonomy of infrastructure systems is defined for the energy domain. It is further specialized in specific sub-domains and instantiated in use case knowledge graphs with specific real-world (XofInterest) entities: MaiaMunici-palityChargingStation, NarbonnePVPlant,...

Domain ontologies are created by ontology engineers for each domain (and sub-domain, in collaboration with domain experts), while data providers handle the knowledge graph creation with instantiation of specific properties and systems in the scope of their use case.

The Omega-X ontology is to be used as metadata source for the data exchange process, and the service providers make use of created knwoledge graphs to enhance their developped services (prediction services, analysis, gamification,...).

This projetc shows that the pattern "*Kind of X and X of Interest*" can be considered beyond web of things projects to ease consideration separations.

## 7. Conclusion

In this paper, we introduce the "Kind of X and X of Interest" ontology design pattern to address the modelling ambiguities and integration challenges identified in standard Web of Things (WoT) ontologies. The pattern conciliates the differences between various design strategies of the SSN/SOSA, WoT Thing Description, and SAREF ontologies regarding specific and generic features, properties, devices... The application of the pattern in the CoSWoT ontology shows a successful separation of considerations of generic types and specific instances, enabling a flexible approach for domain ontology designers and application developers and the alignment with controlled vocabularies and taxonomies.

The paper also discusses a practical use case on smart building applications where an extension of the ontology to the domain is made possible through the X class, the use of SKOS concepts is integrated through XKinds, and instances of XofInterest are created to populate the knowledge graph of the application specific features of interest, properties, devices,...

Moreover, we highlight the potential for generalizing this pattern use beyond WoT applications, as illustrated by its use in the Omega-X project, which addresses energy data interoperability. This motivates the potential use of the pattern "Kind of X and X of Interest" in various domains where separation of consideration in ontology design is required. Further efforts are planned to consider punning integration and automatic extension of class hierarchies for specific domains.

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
