# OpenReview forum: "Kind of X and X of Interest: An Ontology Design Pattern to Reconcile Web of Thing Ontologies"
_swsa.semanticweb.org/ISWC/2024/Workshop/WOP — WOP 2024 Oral_

### Official Review · Reviewer_FGCj · 2024-08-26
**Review of Paper 2: Kind of X and X of Interest: An Ontology Design Pattern to Reconcile Web of Thing Ontologies**

**Rating:** 9
**Confidence:** 3

**Review:**

### Overview:
In this paper, the authors propose a new pattern to harmonize the distinction of generic and specific concepts in Web of Thing (WoT) ontologies, namely the “Kind of X and X of Interest” pattern.  Aiming to bridge the gap between existing best practice ontologies of the domain, they suggest a new framework that can take both perspectives into account and enable a more faceted way of modeling measurements and actions. Next to a case study in the domain of WoT (CoSWoT project), they also showcase an application outside of this scope with Omega-X.

### Practical Utility and Reusability of the Pattern (within a Community):
The pattern proposed by the authors is a valuable asset to the existing approaches (primarily, but not only) in the WoT domain. It captures the core principles of generality and specificity in a unified framework, which developers can use to define a more holistic view on their applications, as exemplified by the CoSWoT and Omega-X use cases. As a meta-pattern, it enables interoperability between different modeling approaches, making it possible to integrate data from multiple sources and providing a common understanding of how to represent properties and features. The proposed “Kind of X and X of Interest” pattern provides clarity in terms of reduced ambiguity for the use of ontological definitions from WoT ontologies in real-world scenarios, making it easier to understand in which scenario an “X” is to be instantiated as generic or specific. As proposed in this paper, the pattern's flexible structure makes it applicable outside of the WoT domain, which enhances its utility and promotes both inter- and intra-domain interoperability.


### Relevance of the Problem Addressed by the Pattern:
The pattern addresses a relevant issue in the WoT/IoT domain. Recognizing that despite the existence of best practices for such applications, interoperability challenges remain, the authors propose this pattern to reconcile the challenge of data integration and communication between devices across different modeling approaches. In particular, it addresses the ambiguities and inconsistencies resulting from a lack of clarity for generic and specific entities in ontologies such as SAREF, TD, and SOSA/SSN. They address the need for convergence between modeling practices, using the pattern to overcome the way in which unclear definitions foster misinterpretations and errors in data representation and integration. This is particularly relevant as IoT applications depend on interoperability, and to enable comprehensive modeling of processes, organizations and projects often share ontologies such as SAREF and SOSA to describe similar concepts. Additionally, the proposition of patterns for the domain suggest modular approaches, which can further reduce complexities of modeling approaches and foster maintainable solutions.


### Best Practice within a Community:
As shown by the consideration of different ontologies in the domain, the pattern proposed by the authors represents a best practice in the WoT community. It is a reconciliation of the core concepts in ontologies such as SAREF and SOSA, and uses existing conceptualizations (such as XKind of XofInterest), merging them into a unified framework to bridge the gap between generic and specific representations.


### Real-World Use:
The authors demonstrate the real-world use of the “Kind of X and X of Interest” pattern through two case studies. For example, the application of this pattern in the CoSWoT project, which aims to extend the Semantic Web of Things (SWoT) to constrained devices, shows how the pattern can be applied not only to properties, but to the full range of conceptualizations relevant to the representation of devices, the physical processes involved, computation, and networking. This successful application highlights the relevance and utility of the pattern for real-world scenarios in the IoT domain. In addition, the authors have included a case study from outside the IoT domain, the Omega-X project, to demonstrate how the pattern can find generalized application outside the IoT domain: Omega-X is a project in the energy sector that aims to create a data space for secure and innovative data exchange, highlighting the importance of interoperability. Here, there is a separation between taxonomic definitions and real-world instantiation in knowledge graphs for properties and infrastructures. Thus, the authors show the utility of the pattern to support the creation of knowledge graphs and the definition of entities and their relationship.


### Overall evaluation, questions, and recommendations:
* How is this pattern ‘different’ from other ontology design patterns (what makes it stand out in the field)? How does the pattern facilitate the creation of knowledge graphs? Would you characterize it as a kind of meta-pattern that can bring benefits to more domains outside of IoT and projects like Omega-X, e.g., the general distinction between A-BOX and T-BOX in knowledge graph construction? Maybe you could add a few more words about this in your Uptake section.
* How do you see the future impact of this pattern for ontologies like SOSA that do not (yet) formally distinguish between generic and specific properties? Would SOSA need new classes to apply this pattern or is your pattern powerful enough to bridge the gap between the SOSA approach and others like SAREF? Could you explain, perhaps with an exemplary modeling of a process in SOSA, SAREF, and one with the pattern, how these merge in real-world KG that may need to exchange information but depend on different modeling choices? Perhaps you could also add, more specifically, which parts of the pattern address which conceptualization in the other ontologies (similar to the figures in section 3 and section 6, showing where the core concepts stem from and how you reconcile them)?

The pattern presented in this paper contributes significantly to the development of ontologies and related conceptualizations in the context of WoT/IoT. It is an innovative approach, based on existing core patterns from best practices in the domain, to bridge the multidimensional representation of devices and implicit generics and specifics at different levels (features, properties, etc.) and bring clarity to the distinction. Thus, it contributes to the field with multi-perspective characteristics of WoT applications, reconciling different ontological approaches (e.g., SAREF, SOSA/SSN), thereby promoting interoperability between different systems and standards, and potentially enhancing cross-device communication in the IoT, which is crucial in a landscape where data sharing and integration are gaining in importance.

In general, this paper is written in a clear and structured manner, which facilitates the understanding of the complex concepts mentioned in the different sections. In addition, the authors have provided numerous visualizations that have helped immensely in understanding the intent behind the pattern and following the development in relation to existing standards.
To further demonstrate how the pattern can support real-world applications, the paper should highlight direct comparisons between ‘traditional’ modeling examples in e.g. SOSA and the new approach with the pattern (as suggested in the second question). This would further emphasize its relevance and applicability.

---

### Official Review · Reviewer_Q7iZ · 2024-08-29
**Review of "Kind of X and X of Interest: An Ontology Design Pattern to Reconcile Web of Thing Ontologies"**

**Rating:** 7
**Confidence:** 4

**Review:**

The authors have proposed an ontology design pattern for aligning and integrating concepts in the web of things, specifically in SOSA/SSN, WoT Thing Descriptions, and SAREF ontology. Each ontology has its unique modeling choices. Entity alignment, a crucial issue in the knowledge graph community, and its use in ontology are the focus of this paper. Some ontologies model an instance of the class as generic, while others model them as a specific entity of interest. The authors demonstrate the pattern using the CoSWoT project use case.

Suggestions:
The word "architectural design pattern" is confusing. Please use "ontology design pattern" instead. Please consider having the figures on the same page as the references. Figure 5 is a page later. Please consider using legends for the figures. Please use diagram tools used in the community instead of Chowlk. It would also be great to define clear competency questions for the use case that can be answered using the pattern (you can refer to papers published in the past for reference).

---

### Official Review · Reviewer_xrbY · 2024-09-04
**Review of "Kind of X and X of Interest: An Ontology Design Pattern to Reconcile Web of Thing Ontologies"**

**Rating:** 8
**Confidence:** 3

**Review:**

## Summary

This paper addresses the difficulty of aligning different Web of Thing (WoT) ontologies, namely, SOSA/SSN, WoT Thing Descriptions, and SAREF. These have similar goals but vary in their modeling approaches. The authors propose a design pattern called "Kind of X and X of Interest," which reconciles these differences by introducing a systematic way to distinguish between generic and specific entities within ontologies. The pattern is demonstrated through its application in the CoSWoT project, which extends the Semantic Web of Things to constrained devices.

## Positive Aspects:
- The paper proposes a new pattern which allows multiple WoT ontologies to be interoperable. This manages to address a significant issue bridging the gap between representation schemes and allowing standardization.
- Provides a demonstrable example of its effectiveness by applying this proposed pattern in the CoSWoT project.
- The paper also discusses the potential for generalization of the pattern and its applicability to other domains.

## Potential Concerns & Weaknesses:
- Given that this is not a short paper, a bit more introductory details to help people understand the work would be helpful. Simpler examples can be illustrative in describing the related concepts and explaining the work.
- Figure 1 could be improved by capturing more details, even if they are described later in the text.
- A broader evaluation of the design pattern beyond CoSWoT would be beneficial.

## Minor Language/Formatting Issues:
- Page 2: "Yet, even them have ambiguities" – they? (rephrase)

## Note:
- Design pattern cannot be found at the community portal specified in the CFP (https://github.com/odpa/patterns-repository).